# Diabetes Aggravates Photoreceptor Pathologies in a Mouse Model for Ocular Vitamin A Deficiency

**DOI:** 10.3390/antiox11061142

**Published:** 2022-06-10

**Authors:** Srinivasagan Ramkumar, Vipul M. Parmar, Jean Moon, Chieh Lee, Patricia R. Taylor, Johannes von Lintig

**Affiliations:** 1Department of Pharmacology, School of Medicine, Case Western Reserve University, Cleveland, OH 44106, USA; rxs787@case.edu (S.R.); vipulkumar.parmar@duke.edu (V.M.P.); jxm1019@case.edu (J.M.); 2Department of Ophthalmology, School of Medicine, Case Western Reserve University, Cleveland, OH 44106, USA; cxl77@case.edu (C.L.); patricia.r.taylor@case.edu (P.R.T.)

**Keywords:** diabetes, vitamin A homeostasis, *Stra6*, oxidative stress, photoreceptors

## Abstract

Emerging evidence indicates that diabetes disturbs photoreceptor function and vitamin A homeostasis. However, the biochemical basis of this phenotype is not well established. Here, we compared the effects of streptozotocin-induced diabetes in wild-type (WT) mice and *Stra6^-/-^* mice, a mouse model for ocular vitamin A deficiency. After 8 weeks, diabetes increased serum retinyl esters in mice of both genotypes. The eyes of diabetic WT mice displayed increased superoxide levels but no changes in retinoid concentrations. Diabetic *Stra6^-/-^* mice showed increased ocular retinoid concentrations, but superoxide levels remained unchanged. After 30 weeks, significant alterations in liver and fat retinoid concentrations were observed in diabetic mice. Diabetic WT mice exhibited a decreased expression of visual cycle proteins and a thinning of the photoreceptor layer. *Stra6^-/-^* mice displayed significantly lower ocular retinoid concentration than WT mice. An altered retinal morphology and a reduced expression of photoreceptor marker genes paralleled these biochemical changes and were more pronounced in the diabetic animals. Taken together, we observed that diabetes altered vitamin A homeostasis in several organ systems and aggravated photoreceptor pathologies in the vitamin-deficient mouse eyes.

## 1. Introduction

Diabetes is a chronic metabolic disease, and a majority of individuals develop microvascular complications such as diabetic retinopathy (DR) within two decades of disease onset [1]. The disease has achieved epidemic proportions and, consequently, DR is a leading cause of blindness and a major societal problem [2]. DR is characterized by damage to the blood vessels at the back of the eye and is typically diagnosed using fundus photography focused on these blood vessels. The disease initially presents with mild visual impairment but can progress to blindness when the pathology proceeds. Early-stage diabetic retinopathy, known as non-proliferative diabetic retinopathy (NPDR), presents with microaneurysms and sometimes edema. Patients with NPDR do not display vision loss, but the disease can progress to proliferative diabetic retinopathy (PDR), with characteristic neovascularization and the potential for vascular hemorrhage. This pathology results in a blinding disease which presents as patchy vision. While the exact mechanism of the pathology of DR is still under investigation [3], some investigators suggest that the photoreceptor and visual (retinoid) cycle associated with oxidative stress from photoreceptors in the retina may play a significant role in the etiology of the disease [4].

The visual cycle is the enzymatic pathway through which visual chromophore is recycled after the cone and rod visual pigments are bleached [5]. The enzyme-catalyzed process takes place between photoreceptors and the retina pigment epithelium (RPE) [6]. The RPE performs major steps in chromophore regeneration and acquires vitamin A from the circulation [7]. In the circulation, vitamin A exists bound to the serum retinol-binding protein (holo-RBP4) and as retinyl esters (RE) in chylomicrons [8]. The cellular uptake of vitamin A from these two transport modes of the vitamin are facilitated, by the RBP4 receptor STRA6 (stimulated by retinoic acid 6) and lipoprotein lipase, respectively [9,10,11]. STRA6 is essential for ocular vitamin A homeostasis and, in its absence, cone and rod photoreceptor functions are compromised [12,13,14].

Evidence from the clinical and experimental animal suggests that photoreceptor function and vitamin A metabolism are significantly altered in diabetes [15]. Patients with type 1 diabetes (T1D) display low serum RBP4 levels [16], and serum retinol is directly proportional to RBP4 in these patients and is also lowered [17]. The photoreceptors of the retina are a major destination for the delivery of RBP4′s cargo, and the involvement of this neuronal cell type in the pathogenesis of DR has recently been indicated [4,18,19,20]. Clinical studies indicate that rod sensitivity is subnormal in early DR with abnormalities in dark adaptation and absolute threshold [4]. Studies in rodents have revealed that ocular retinoid homeostasis and visual pigment metabolism are altered in diabetes [21]. Accordingly, the treatment of diabetic rodents with the chromophore surrogate 9-*cis*-retinal ameliorates oxidative stress and improves photoreceptor function [22,23]. These findings suggest that diabetes alters ocular vitamin A homeostasis and affects photoreceptor function.

However, the role of vitamin A metabolism in diabetes is still understudied. Therefore, we compared the short-term and long-term consequences of diabetes in wild-type (WT) mice and *Stra6^-/-^* mice, a model for ocular vitamin A deficiency [24]. *Stra6^-/-^* mice display ocular vitamin A deficiency and unliganded opsin even on vitamin A rich diets [12], thus allowing us to study the interaction of diabetes and vitamin A deficiency in the eyes of these mice.

## 2. Materials and Methods

### 2.1. Animals, Husbandry, and Diabetic Induction

All procedures involving animals were performed in accordance with the Association for Research in Vision and Ophthalmology (ARVO) Statement for the Use of Animals in Ophthalmic and Vision Research and were reviewed and approved by the Case Western Reserve University IACUC. Male C57BL/6J mice were obtained from the Jackson Laboratory (Bar Harbor, ME, USA). The generation of *Stra6^-/-^* mice was reported earlier [14]. All mice were housed with free access to food and water. Diabetes was induced in 8–10-week-old male mice by intraperitoneal injections of streptozotocin (STZ) (MPBio, Solon, OH, USA) at 60 mg/kg bodyweight for 5 consecutive days. Diabetes was defined by 6 h fasted blood glucose concentrations greater than 275 mg/dL, which was verified using glucose-dehydrogenase-based strips 17 days after the last STZ injection (Day 22). Hyperglycemia was quantified by hemoglobin A1c levels using the Crystal Chem (Elk Grove Village, IL, USA) Mouse A1c kit at weeks 6, 16, and 30 after diabetes was confirmed. The procedure was performed according to the manufacturer’s protocol. Insulin (Humulin N, Eli Lilly, Indianapolis, IN, USA) at 0–0.2 U was administered as needed to maintain proper body weight.

### 2.2. Tissue Isolation

Eyes were enucleated based on the methods described in [25]. Briefly, mice were euthanized by rodent cocktail (20 mg/mL Ketamine + 1.75 mg/mL Xylazine). Blood was taken to generate serum and mice were perfused by phosphate-buffered saline (PBS). The eyes were gently proptosed, and curved forceps (with the curve facing upwards) were used to apply gentle pressure to the temporal side of the orbit to further proptose the eye, allowing the forceps to straddle the globe until they could clamp onto the optic nerve. A quick motion was then used to enucleate the eye. Other tissues such as epididymal white adipose tissue (eWAT) and the liver were dissected with surgical scissors, snap frozen in liquid nitrogen, and stored at −80 °C until further use.

### 2.3. Retina Isolation from Enucleated Eyes for ROS Assays

The enucleated eye was placed onto a piece of dental wax and held in place via the remnants of the extraocular muscles while a Teflon-coated razor blade was used to cut off the anterior eye at the cornea–sclera junction. The retina was then gently detached from the posterior eyecup in preparation for immediate use in the assays described below.

### 2.4. Quantification of Reactive Oxygen Species (ROS)

Individual retinas were isolated and incubated in Krebs-HEPES buffer (with 5 mmol/L glucose) for 25 min at 37 °C in 5% CO_2_. Luminescence was measured (using Promega GLOMAX 20/20 luminometer) 5 min after the addition of 0.5 mmol/L lucigenin, as previously described [18,26], to quantify the level of ROS per retina.

### 2.5. HPLC Analysis

To each eye, 200 µL of 2M Hydroxylamine (pH 6.8) was added, homogenized, and sonicated for 30 s. Retinoids were extracted with a mixture containing 200 µL methanol, 400 µL acetone, and 500 µL hexane. Retinoids were also extracted from 100 µL of serum, 10 mg of liver, and 50 mg of eWAT by mixing with 200 µL PBS and 200 µL methanol. Then, 400 µL acetone and 500 µL hexane were added and the organic phases were collected and dried in a Speedvac (Eppendorf, Hamburg, Germany). HPLC analyses were performed on a normal-phase Zorbax Sil (5 μm, 4.6 × 150 mm) column. Chromatographic separation was achieved by the isocratic flow (1.4 mL/min) of 90% hexane with 10% ethyl acetate. HPLC has previously been scaled with synthesized standards for the quantification of retinoid molar amounts.

### 2.6. SD-OCT and Fundus Imaging

The pupils of *Stra6**^-/-^* and WT mice were fully dilated with 1% tropicamide (Falcon Pharmaceuticals, Fort Worth, TX, USA) and mice were anesthetized by an intraperitoneal injection of rodent anesthetic cocktail (20 mg/mL ketamine and 1.75 mg/mL xylazine). Mice whiskers were trimmed to avoid artifacts. Spectral domain optical coherence tomography (SD-OCT) images were acquired in the linear B-scan mode of an ultra-high-resolution SD-OCT instrument (Bioptigen, Morrisville, NC, USA). Confocal scanning laser ophthalmoscopy (SLO; Spectralis HRA2; Heidelberg Engineering, Heidelberg, Germany) was performed with a 55-degree lens to collect mouse fundus images. The near-infrared (IR) reflectance image (IR mode, 820 nm laser) was used to align the fundus camera relative to the pupil to obtain an evenly illuminated fundus image. ONL thickness was calculated using the software of the SD-OCT instrument. The thickness of the ONL was measured at a 0.2 mm distance from the optic nerve head and measurements were reported at every 0.1 mm interval.

### 2.7. RNA Extraction and q-RT-PCR Analysis

RNA was extracted from dissected retina and liver by the TRIZOL method (Invitrogen, Carlsbad, CA). Total RNA was quantified using a Nanodrop ND-1000 spectrophotometer (Thermo Fisher Scientific, Waltham, MA, USA). cDNA was generated using the high-capacity RNA to cDNA kit (Applied Biosystem; Thermo Fisher Scientific, Waltham, MA, USA). Gene expression measurement was carried out by real-time quantitative PCR (qPCR) using the Applied Biosystem Real-Time PCR instrument with Taq Man probes (Applied Biosystem; Thermo Fisher Scientific, Waltham, MA, USA), primer *β-actin* (Mm02619580), *Gnat1* (Mm01229120), *Gnat2* (Mm00492394), *Rho* (Mm01184405), *Opn1mw* (Mm00433560), *Opn1sw* (Mm00432058), *Lrat* (Mm00469972), *Rbp1* (Mm00441119), *Cyp26a1* (Mm00514486), *Rarb* (Mm01319677), and *Rbp4* (Mm00803266). Amplification was carried out using TaqMan Fast Universal PCR Master Mix (2×) No AmpErase UNG (Applied Biosystem; Thermo Fisher Scientific, Waltham, MA, USA) following the manufacturer’s protocol. An amount of 10 ng cDNA was used per 10 µL reaction mixture. Gene expression levels were normalized to expression using the ∆∆C_t_ method. Statistical significance was calculated using one-way ANOVA by graph pad prism 8 software(GraphPad Software, San Diego, CA, USA) and the results were considered significant at * *p* < 0.05, ** *p* ≤ 0.001, and *** *p* ≤ 0.0001.

### 2.8. Western Blot Analysis

For Western blot analysis, eye cups consisting of RPE, choroid and sclera were harvested at 30 weeks from STZ-induced diabetic and non-diabetic WT mice. The isolated eye cups were homogenized in RIPA lysis buffer consisting of Complete Mini EDTA-free protease inhibitor (Roche Applied Science, Penzberg, Germany). The protein content was measured using the BCA protein assay. An equal amount of protein for each condition (10–20 ug) was loaded in 12% SDS-PAGE gel. Proteins were transferred to the PVDF membrane through an electroblot. Proteins were detected by Western blotting with the specific primary antibody. After proteins were transferred to the membranes, they were blocked with fat-free milk powder (5%, *w*/*v*) dissolved in Tris-buffered saline (15 mM NaCl and 10 mM Tris/HCl, pH 7.5) containing 0.1% Tween 20 (TBS-T), washed, and incubated overnight at 4 °C with the appropriate primary antibody. For CRBP1 detection, a polyclonal rabbit antibody (Santa Cruz Biotechnology, Inc., Santa Cruz, CA, USA) was employed at a 1:1000 dilution. For LRAT detection, a non-commercial in-house-developed anti-LRAT monoclonal antibody was used at a dilution of 1:2000. For RPE65 detection, a non-commercial in-house-developed RPE65 antibody was used at a dilution of 1:2000. β-actin was used as a loading control, and for β-actin detection antiserum for β-actin (Cell Signaling, Boston, MA, USA) was used at a dilution of 1:5000. The secondary antibodies employed were either horseradish peroxidase-conjugated anti-rabbit IgG (Promega, Madison, WI) or anti-mouse IgG (Promega, Madison, WI, USA) used at a dilution of 1:5000. Immunoblots were developed with the ECL system (Thermo Fischer Scientific, Waltham, MA, USA).

### 2.9. Statics Analysis

All results were expressed as means ± SD. The data were analyzed by one-way ANOVA by using graph pad prism 8 software. Data were considered statistically significant when * *p* < 0.05, ** *p* ≤ 0.001, and *** *p* ≤ 0.0001.

## 3. Results

### 3.1. Design of the Mouse Studies

We took advantage of an established STRA6-deficient mouse line and isogenic WT mice [14]. Mice were bred on vitamin A-rich breeder chow. Groups of mice (diabetic and non-diabetic WT and *Stra6^-/-^* mice) had equal numbers (*n* = 12) and were of the same C57BL/6J genetic background. We used male mice in the study because of sex-specific differences in vitamin A metabolism and DR. Once mice reached 8 weeks of age, they were injected with streptozotocin for 5 consecutive days, damaging pancreatic β cells and inducing T1D (Figure 1). Age- and sex-matched non-treated mice served as controls for both genotypes. After 8 weeks of diabetes, mice of the first cohort (*n* = 6 per genotype and treatment group) were sacrificed in the morning, since circadian rhythmicity influences several key physiological parameters in diabetes. After 30 weeks, the second cohort of mice was sacrificed (*n* = 6 per genotype and treatment group) to determine the long-term effects of T1D on vitamin A homeostasis and ocular health in the different genotypes.

### 3.2. Short-Term Effects of Diabetes on Retinoid Metabolism

After 8 weeks of diabetes, we determined the levels of Hemoglobin A1c (HbA1c) in the blood of the mice by an established ELISA method. HbA1c results from the non-enzymatic modification of the N-terminal amino acid of the hemoglobin molecule by hexose sugars, and its occurrence is dependent on blood glucose concentration. Diabetic mice displayed a significant increase in HbA1c in their blood samples when compared to non-diabetic mice (Figure 2A). This outcome confirmed that we successfully induced diabetes in *Stra6^-/-^* and WT mice. Body weights were slightly but significantly lower in diabetic mice of both genotypes (Figure 2B). We then sacrificed half of the mice for more detailed analysis. Liver weights were increased in diabetic *Stra6^-/-^* when compared to non-diabetic littermates and the same trend was noticed in WT mice, but this did not reach significance (Figure 2C). Retinol blood concentrations were in the range of 0.6 μM, and did not show a significant difference between treatment groups and genotypes at this early stage of diabetes (Figure 2D). Remarkably, we detected relatively high concentrations of retinyl esters in the serum of the diabetic mice of both genotypes, whereas the ester form of the vitamin was largely absent in the non-diabetic littermates (Figure 2E). Hepatic ROL levels were increased in diabetic mice over the levels of the non-diabetic littermates (Figure 2F). Hepatic RE levels were comparable between genotypes and treatment groups (Figure 2G). Taken together, streptozotocin injections induced diabetes in mice as indicated by the significantly elevated HbA1c levels. Diabetic mice displayed significant alteration in retinoid concentrations in non-ocular tissues, including increased serum RE levels.

### 3.3. Ocular Retinoid Homeostasis and Oxidative Stress in WT and Stra6^-/-^ Mice

We next determined ocular retinoids by HPLC analysis. In both genotypes and treatment groups, we detected all intermediates of the visual cycle (Figure 3A,B). The eyes of *Stra6^-/-^* mice contained significantly lower concentrations of retinoid than that of WT control mice in both treatment groups (Figure 3C). This observation confirms the critical role of STRA6 in acquiring vitamins from the circulation to satisfy the ocular needs of chromophore synthesis [12,13,24]. We observed no effect of diabetes on total ocular retinoid concentration in WT mice (Figure 3C). Remarkably, diabetic *Stra6^-/-^* mice displayed a significant increase in total ocular retinoid concentration when compared to their non-diabetic littermates (Figure 3C). This increase in ocular retinoid concentration likely reflects the high serum RE levels in diabetic *Stra6^-/-^* mice. REs in lipoproteins are the major source of vitamin A for ocular tissues in the absence of the RBP receptor [24].

An established early indicator of DR pathogenesis is retinal oxidative stress. We evaluated oxidative stress after 8 weeks of diabetes by measuring superoxide generation from freshly isolated retinas and compared it to non-diabetic mice of both genotypes. For this purpose, luminescence indicating the presence of superoxide was measured after the addition of lucigenin to retina homogenate using an established protocol. Superoxide production was significantly increased in diabetic WT mice when compared to non-diabetic WT mice (Figure 3D). Remarkably, diabetic and non-diabetic *Stra6^-/-^* mice showed relatively high superoxide concentrations in this experiment (Figure 3D), indicating that the *Stra6* genotype was associated with oxidative retinal stress. Taken together, we observed that diabetes had no measurable effect on ocular retinoid homeostasis in WT mice. In *Stra6^-/-^* mice, diabetes increased total ocular retinoid concentration but not oxidative stress in the retina when compared to non-diabetic *Stra6^-/-^* mice.

### 3.4. Long-Term Effects of Diabetes on Retinoid Metabolism

To confirm diabetes in WT and *Stra6^-/-^* mice after 30 weeks, we again determined the HbA1c concentration in the serum of the remaining mice. The analysis revealed that HbA1c serum concentrations had continued to rise in diabetic mice and were more than 2-fold higher than in the non-diabetic mice of both genotypes (Figure 4A). The diabetic mice of both genotypes displayed significantly lower total body weights when compared to the non-diabetic controls (Figure 4B). The liver weights of the diabetic and non-diabetic mice were comparable (Figure 4C); however, the fat mass of diabetic animals was highly reduced as indicated by the analysis of the epididymal adipose depots (Figure 4D). The HPLC analysis of fat showed that the concentration of ROL and RE had increased in the diabetic mice. However, the overall retinoid content of the fat depots was lower because of their reduced masses (Figure 4E,F). HPLC analysis showed that serum retinol levels were comparable between diabetic and non-diabetic mice but tended to be higher in *Stra6^-/-^* mice when compared to WT mice (Figure 4G). Liver ROL levels were significantly increased in diabetic mice when compared to non-diabetic mice. This increase was more pronounced in *Stra6^-/-^* mice (Figure 4H). A similar trend was observed for hepatic RE levels, which were higher in *Stra6^-/-^* mice than in WT mice (Figure 4I). qRT-PCR revealed that the expression of the *Rbp1* gene encoding cellular retinol-binding protein was increased in diabetic mice, consistent with their increased hepatic ROL levels (Figure 5A). The expression of *Lrat* and *Rbp4* mRNA, encoding the major vitamin A esterifying enzyme and the serum retinol-binding protein, respectively, were unchanged (Figure 5B,C). The mRNA expression levels of the genes encoding the retinoic acid catabolizing *Cyp26a1* and the retinoic acid receptor β (*Rarb*) were significantly increased in the liver of diabetic mice (Figure 5D,E), indicating that diabetic mice showed enhanced retinoid signaling in this organ.

### 3.5. Ocular Consequences of Long-Term Diabetes

The determination of ocular retinoid concentrations by HPLC demonstrated that all retinoid cycle intermediates existed in *Stra6^-/-^* and WT mice (Figure 6A,B). After 30 weeks, diabetic and non-diabetic *Stra6^-/-^* mice still displayed a significantly lower concentration of ocular retinoids when compared to WT mice (Figure 6C). This finding confirmed previous results that showed that STRA6-deficient eyes cannot reach the retinoid concentration of WT mice even when continuously kept on diets with high vitamin A supplies [12]. In contrast to the measurements after 8 weeks, diabetic and non-diabetic *Stra6^-/-^* mice showed comparable retinoid concentration in their eyes at 30 weeks (Figure 6C). Diabetic WT mice displayed slightly lower ocular retinoid concentrations than non-diabetic littermates, but this difference did not reach significance. We also determined the expression levels of the key enzymes of the visual cycle in diabetic and non-diabetic WT mice. As shown in Figure 7, the protein concentrations of RPE65 and CRBP1 were comparable in the RPE between the treatment groups. However, LRAT expression was apparently lower in diabetic than in non-diabetic WT mice.

Rod and cone photoreceptors express specific opsins, rhodopsin (encoded by the *Rho* gene) in rods, and M-opsin (encoded by the *Opn1mw* gene) and S-opsin (encoded by the *Opn1sw* gene) in the two cone photoreceptor types [27]. We next determined the expression levels of the respective genes in the eyes of the diabetic and non-diabetic *Stra6^-/-^* and WT mice. We observed that the expression of the *Rho* mRNA in *Stra6^-/-^* diabetic mice was significantly reduced compared to non-diabetic *Stra6^-/-^* mice (Figure 6D). Diabetic and non-diabetic WT mice did not display such a difference in *Rho* expression in their retinas (Figure 6D). The expression levels of *Opn1mw* and *Opn1sw* were lower in *Stra6^-/-^* mice than in WT mice (Figure 6E,F). mRNA levels of *Opn1mw* were further decreased in diabetic mice when compared to non-diabetic *Stra6^-/-^* mice (Figure 6E). However, there was no difference in *Opn1sw* expression between diabetic and non-diabetic mice (Figure 6F). Additionally, we analyzed the expression of the G protein subunits alpha transducin 1 (*Gnat1*) and transducin 2 (*Gnat2*), which are rod- and cone-specific components of phototransduction machinery (Figure 6G,H). There was a significant difference in the expression of *Gnat1* mRNA in the retinas of non-diabetic and diabetic *Stra6^-/-^* mice, but not in non-diabetic and diabetic WT mice. (Figure 6). The expression of *Gnat2* followed a similar pattern as the expression of the *Opn1mw* gene being reduced in diabetic *Stra6^-/-^* mice (Figure 6H).

Diabetes can trigger retinal pathologies such as changes in retinal vasculature as well as retinal thinning. Thus, we examined the retinal morphology of diabetic and non-diabetic *Stra6^-/-^* and WT mice using optical coherence tomography (OCT) and scanning laser ophthalmoscopy (SLO) (Figure 8A–L). The OCT analysis revealed that all mice had a normal stratification of retinal layers. The diabetic and non-diabetic *Stra6^-/-^* mice displayed a reduced thickness in their outer nuclear layer (Figure 8G,H) indicative of a reduction in the number of rod and cone photoreceptors. To quantify the thinning of the outer nuclear layer, the averaged thicknesses of different parts of the retina were plotted in a graph (Figure 8M). This analysis showed that the thinning was found in both inferior and superior parts of the retina of diabetic and non-diabetic *Stra6^-/-^* mice (Figure 8M). A thinning of the outer nuclear layer was also observed in the inferior part of the retina of diabetic WT mice, though this was less pronounced than in *Stra6^-/-^* mice (Figure 8M). The fundus fluorescein angiography of diabetic and non-diabetic mice did not reveal pronounced damage of the vasculature, neither in diabetic mice nor in *Stra6^-/-^* mice (Figure 8C,F,I,L). The latter finding is in accordance with other reports that did not observe such pathology in the retinas of streptozotocin-induced diabetic mouse models [28].

## 4. Discussion

Emerging evidence suggests that photoreceptors contribute to the pathology of DR [15]. These specialized neurons are a major site of vitamin A metabolism. To analyze a putative connection between vitamin A metabolism and diabetes, we examined the changes enforced by diabetes on this metabolism in WT mice and *Stra6^-/-^* mice. Our study revealed that diabetes affected retinoid homeostasis systemically in several organs and aggravated the ocular consequences of vitamin A deficiency in the long term. The implications of these findings for the treatment of DR are discussed below.

We induced diabetes by streptozotocin treatment and confirmed the manifestation of the disease by measuring Hemoglobin A1c. The major effects of short-term diabetes on retinoid metabolism included increased hepatic all-*trans*-retinol concentrations and elevated serum retinyl ester levels. In the eyes of WT mice, we observed no significant alteration of retinoid concentration after 8 weeks of diabetes when compared to non-diabetic WT controls. Retinoid concentrations of *Stra6^-/-^* mice were only 50% of those of the age-matched WT mice, confirming that vitamin A uptake homeostasis was impaired in this mouse mutant [13,14]. Remarkably, short-term diabetes had a positive effect on the ocular retinoid phenotype, as seen by a slight but significant increase in the retinoid concentration in diabetic *Stra6^-/-^* mice. The elevated retinoid concentrations in the eyes of the diabetic *Stra6^-/-^* mice are likely explained by the effects of diabetes on lipoprotein metabolism [29]. Retinyl esters in lipoproteins are a source of vitamin A, and were elevated in the blood of diabetic STRA6-deficient mice [24].

Oxidative stress is a hallmark of diabetic retinopathy and one of the first measurable pathologies in diabetic mouse models [15]. A causal relation between oxidative stress and capillary damage in diabetes has been established, since treatment with antioxidants or the overexpression of anti-oxidant enzymes can ameliorate the pathology of the disease. Furthermore, it has been shown that photoreceptors are major contributors to oxidative stress in early diabetes [19,20]. After 8 weeks, diabetic WT mice showed a significant increase in superoxide levels in the retina when compared to non-diabetic littermates. Remarkably, superoxide levels were relatively high in *Stra6^-/-^* mice independent of the diabetic phenotype. The relatively high oxidative stress in the retinas of the *Stra6^-/-^* mice is likely explained by the perturbed visual pigment metabolism in the eyes of these mice. We previously observed that the photoreceptors of *Stra6^-/-^* mice contained significant amounts of unliganded opsins because of insufficient vitamin A supply and chromophore production [12]. Unliganded rod opsin activates sensory transduction in the dark and triggers a slow degeneration of rod photoreceptors [30,31]. Additionally, in the absence of the chromophore, cone opsins and other components of the signaling transduction cascade are not trafficked to the outer segments of the cone photoreceptors [32]. Noteworthily, previous studies have shown that 9-*cis*-retinal supplementation increases the levels of liganded opsins and improves the pathology of photoreceptors in *Stra6^-/-^* mice [12]. Similarly, beneficial effects of 9-cis-retinal supplementation were also reported in diabetic WT mice [12,22,23]. These findings indicate that even minute amounts of unliganded opsins can contribute to oxidative stress in the retina of diabetic WT mice. Nevertheless, we do not want to overlook the notion that additional factors contribute to oxidative stress in the retina in diabetes.

It has been well established that long-term diabetes has a pronounced effect on metabolism. Accordingly, the diabetic mice had significantly lower body weights after 30 weeks of diabetes, with highly reduced body fat stores. As exemplified in eWAT, we observed a higher vitamin A concentration but a lower overall vitamin A content in fat. In the liver, the major place of vitamin A storage, the retinyl ester concentration was higher in diabetic mice. Additionally, all-*trans*-retinol levels were increased in the liver of diabetic mice when compared to the non-diabetic animals. It is meaningful that the hepatic vitamin A concentrations were relatively high in all mice because they were constantly maintained on vitamin A-rich diet. The expression of marker genes for retinoid signaling showed an increased expression of the retinoic acid responsive genes *Cyp26a1* and *Rarb*. Enhanced retinoid signaling has previously been reported in the liver of diabetic mice [33]. There was no significant difference in blood vitamin A levels, though there was a tendency for the elevation of all-*trans*-retinol levels in *Stra6^-/-^* mice. This finding was surprising because some studies have reported that blood vitamin A levels decline in T1D rodents [16,17]. An explanation for the absence of this decline in our study might be the high vitamin A supply and the use of mice instead rats.

In the eyes, a slight reduction in overall retinoid concentrations in diabetic WT mice was observed that did not reach significance. This phenotype was accompanied by a thinning of the outer nuclear layer in the inferior parts of the retina. Angiography revealed no retinal leakage, consistent with the results of previous studies in streptozotocin-induced diabetic mice. The determination of the concentration of proteins that determined vitamin A uptake and metabolism revealed a reduction in the LRAT enzyme that acts downstream of STRA6 in chromophore production [34,35]. Alterations in ocular vitamin A uptake homeostasis and chromophore metabolism have previously been reported in diabetic rats [21,23]. Notably, previous studies have shown that rats are more susceptible to disturbances in retinoid metabolism than mice [36], and this may explain the relatively mild ocular retinoid phenotype of diabetic WT mice in our study. An involvement of retinoid metabolism in the etiology of DR in mice is supported by the finding that retinylamine, a drug targeting ocular retinoid metabolism, ameliorates the ocular consequences of diabetes in mice [37]. The association between diabetes and vitamin A deficiency is further corroborated by our studies in *Stra6^-/-^* mice. The reduced ocular retinoid concentrations were accompanied by an unfavorable photoreceptor physiology that resulted in a significant thinning of the outer nuclear layer. We observed no gross differences in the morphology of the retina between diabetic and non-diabetic animals, indicating that the pathology was mainly driven by the *Stra6* genotype. However, a comparison of the expression of marker genes of photoreceptors indicated a significant deterioration in their functioning in diabetic *Stra6^-/-^* mice. The mRNA levels of *Rho* and *Gnat1,* as well as *Opn1mw* and *Gnat2,* were significantly lower in the retinas of the diabetic mice when compared to the non-diabetic mice. These findings indicated that diabetes aggravated both rod and cone pathologies in this mouse model for vitamin A deficiency.

## 5. Conclusions

In conclusion, our studies in WT and *Stra6^-/-^* mice support the association between diabetes and vitamin A metabolism. As we showed in this study, diabetes altered retinoid homeostasis in several organ systems, including the eyes, blood, liver, and fat. We observed that impaired ocular vitamin A uptake homeostasis aggravated the ocular consequences of diabetes. Clinical studies indicate that patients suffering from T1D have decreased levels of circulating vitamin A and display signs of ocular vitamin A deficiency. Therefore, our study outcome warrants the control of blood vitamin A levels in diabetic subjects and indicates vitamin A supplementation as a therapeutic strategy to ameliorate the symptoms of DR in patients.

## Figures and Tables

**Figure 1 antioxidants-11-01142-f001:**
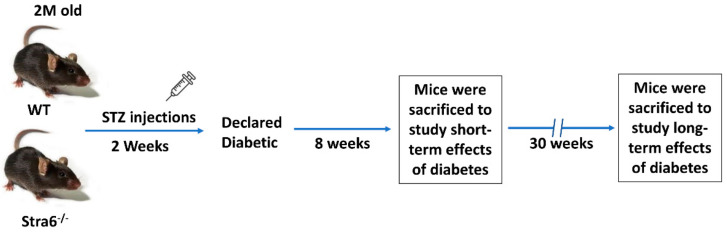
**Experimental setup of the intervention study.** Two-month-old WT and *Stra6**^-/-^**^-/-^* mice were injected with STZ for five consecutive days. Diabetes was declared after the measurement of fasting blood glucose and HBA1c. Eight weeks post-diabetes, mice were sacrificed for short-term experiments. The remaining mice were monitored for an additional 22 weeks and analyzed after 30 weeks of diabetes. In all experiments, equal numbers of non-diabetic mice served as controls.

**Figure 2 antioxidants-11-01142-f002:**
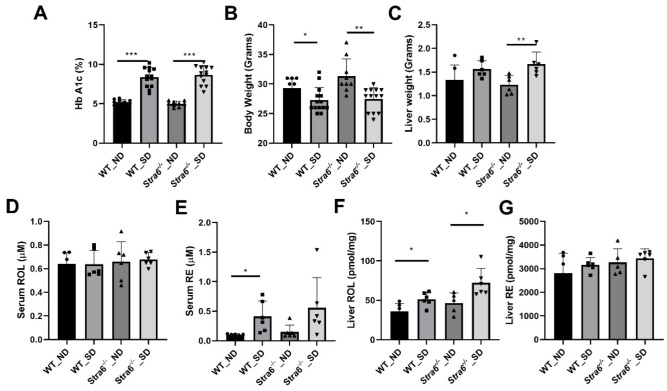
**Effect of short-term diabetes on serum and liver retinoid concentrations**. (**A**) Blood concentrations of hemoglobin A1c. (**B**) Bodyweight. (**C**) Liver weight. (**D**) Serum retinol concentrations. (**E**) Serum retinyl ester concentrations. (**F**) Hepatic retinol concentrations. (**G**) Hepatic retinyl ester concentrations in non-diabetic (ND) and diabetic (SD) WT and *Stra6**^-/-^**^-/-^* mice. The values represent means ± SD of data from at least six animals per genotype and condition. The statistical analysis was performed using a two-tailed student’s *t*-test. * *p* ≤ 0.05, ** *p* ≤ 0.01, *** *p* ≤ 0.001.

**Figure 3 antioxidants-11-01142-f003:**
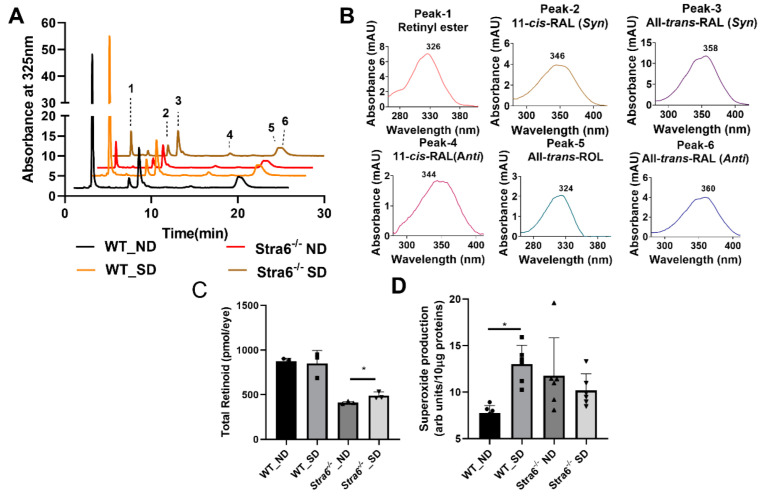
Ocular retinoids and oxidative stress in short-term diabetic and non-diabetic WT and *Stra6**^-/-^* mice. (**A**) HPLC trace at 325 nm of ocular lipid extracts of diabetic and non-diabetic WT and *Stra6^-/-^* mice. Note that the retinal diastereomers were converted to the corresponding retinal oximes and exist as *syn* and *anti* stereoisomers. (**B**) Spectral character of individual ocular retinoids. (**C**) Total ocular retinoids of diabetic and nondiabetic WT and *Stra6^-/-^* mice. (**D**) Superoxide levels in freshly isolated retinal tissue using the lucigenin method from diabetic (SD) and nondiabetic (ND) WT and *Stra6^-/-^.* Values from panel (**C**) represent values from at least three eyes and values in (**D**) represent values from at least six retinas per genotype and condition. The values are expressed as means ± SD. The statistical analysis was performed using a two-tailed *t*-test. * *p* ≤ 0.05.

**Figure 4 antioxidants-11-01142-f004:**
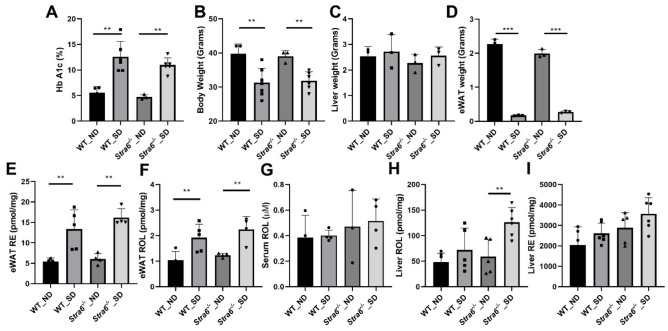
**Long-term diabetes alters retinoid metabolism**. (**A**) Blood hemoglobin A1c concentrations. (**B**) Bodyweight. (**C**) Liver weight. (**D**) eWAT weight. Serum, liver, and eWAT retinoids were measured by HPLC analysis. (**E**) eWAT retinyl ester (RE) concentrations. (**F**) eWAT retinol (ROL) concentrations. (**G**) Serum retinol (ROL) concentrations. (**H**) Hepatic retinol (ROL) concentration. (**I**) Hepatic retinyl ester (RE) concentration levels in non-diabetic (ND) and diabetic (SD) WT and *Stra6^-/-^* mice. The panel (**A**–**I**) values represent means ± SD of data from four to five animals per genotype and condition. The statistical analysis was performed using a two-tailed *t*-test. ** *p* ≤ 0.01 and *** *p* ≤ 0.001.

**Figure 5 antioxidants-11-01142-f005:**
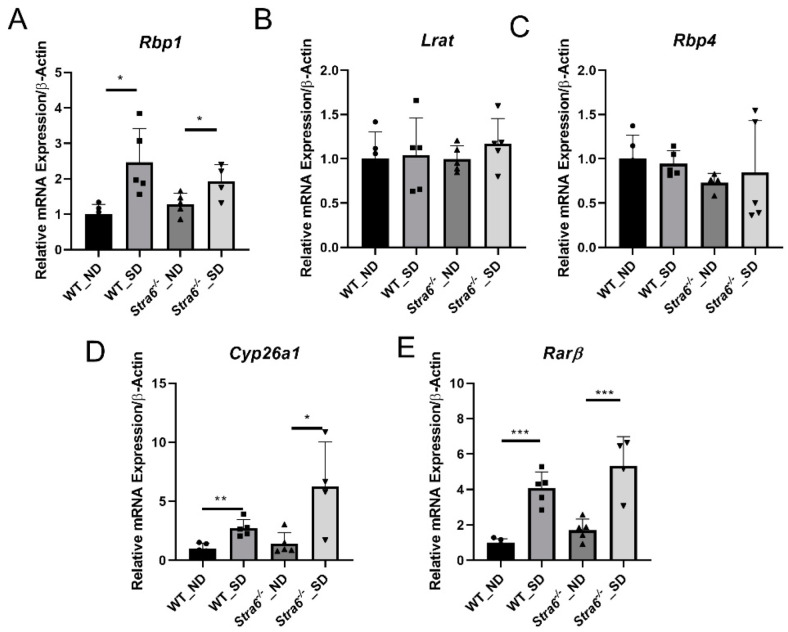
**Diabetes affects hepatic retinoid signaling**. Quantitative RT-PCR analysis for *Rbp1*, *Lrat*, *Rbp4*, *Cyps6a1*, and *Rarβ* mRNA expression levels was performed for the total RNA preparation of the liver of long-term diabetic and non-diabetic WT and *Stra6^-/-^* mice. (**A**) *Rbp1* mRNA level. (**B**) *Lrat* mRNA level. (**C**) *Rbp4* mRNA level. (**D**) *Cyp26a1* mRNA level. (**E**) *Rarβ* mRNA level in non-diabetic (ND) and diabetic (SD) WT and *Stra6^-/-^* mice. β-actin was used as an internal control. The data are represented as means ± SD from at least five animals per genotype and condition. The statistical analysis was performed using a two-tailed *t*-test analysis. * *p* < 0.05, ** *p* < 0.005, and *** *p* < 0.0001.

**Figure 6 antioxidants-11-01142-f006:**
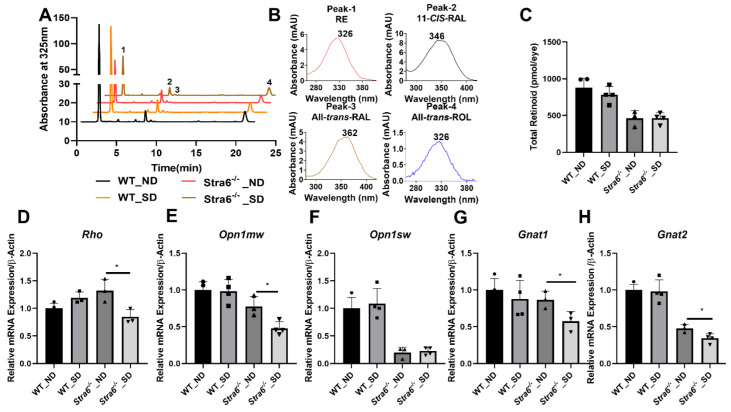
**Ocular retinoid level and gene expression for rod and cone opsins.** (**A**) HPLC trace at 325nm of ocular lipid extracts of long-term diabetic and non-diabetic WT and *Stra6^-/-^* mice. Note that the retinal diastereomers were converted to the corresponding retinal oximes and exist as *syn* and *anti* stereoisomers. (**B**) Spectral characteristics of individual retinoids: RAL, retinaldehyde, RE, retinyl ester, ROL, retinol. (**C**) Total ocular retinoid concentrations. Quantitative RT-PCR analysis of genes of phototransduction machinery. (**D**) *Rho* mRNA levels. (**E**) *Opn1mw* mRNA levels. (**F**) *Opn1sw* mRNA levels. (**G**) *Gnat1* mRNA levels. (**H**) *Gnat2* mRNA levels in non-diabetic (ND) and diabetic (SD) WT and *Stra6^-/-^* mice. *β-Actin* gene was used as an internal control. Values were normalized to the corresponding values of age-matched WT mice and are displayed as mean ± SD of three to four eyes per genotype and condition. The statistical analysis was performed using an unpaired two-tailed student’s *t*-test. * *p* < 0.05.

**Figure 7 antioxidants-11-01142-f007:**
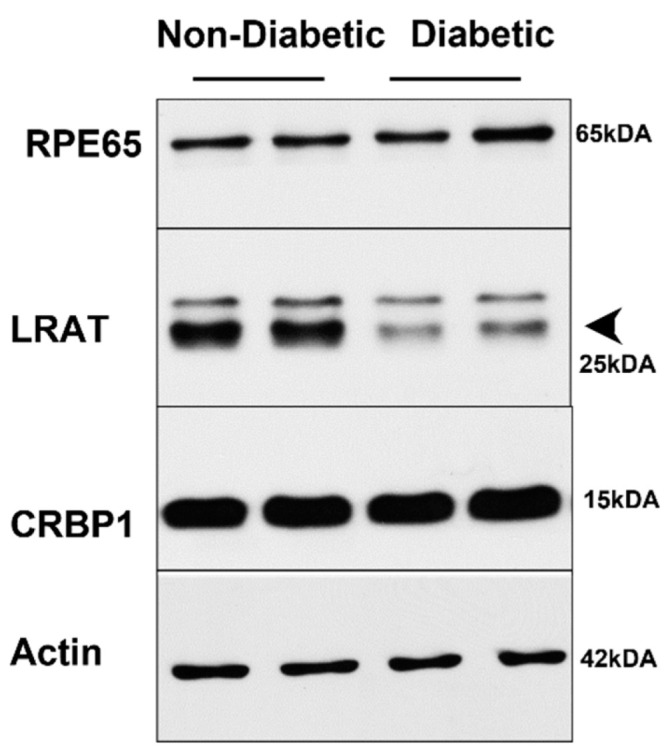
**Western Blot analysis of visual cycle proteins in long-term diabetic WT mice**. Protein extracts from eye cups (30 μg per lane) were separated on SDS PAGE and blotted on membranes as described in the materials and methods section. Representative blots are shown for RPE65, CRBP1, and LRAT proteins.

**Figure 8 antioxidants-11-01142-f008:**
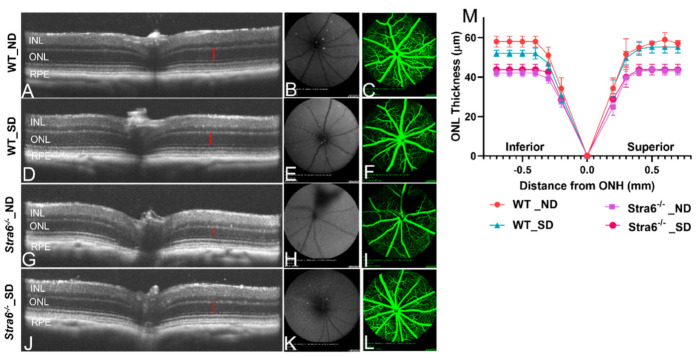
**Imaging analysis of the retinal morphology of long-term diabetic mice**. (**A**–**L**) Representative OCT images (left), fundus photography (middle), and fluorescence SLO (right panel) of long-term (30 weeks) non-diabetic and diabetic WT and *Stra6^-/-^* mice. (**A**) Representative OCT mage for non-diabetic WT mice. (**B**,**C**) Fundus and fluorescence images for non-diabetic WT mice. (**D**) Representative OCT mage for diabetic WT mice. (**E**,**F**) Fundus and fluorescence images for diabetic WT mice. (**G**) Representative OCT mage for non-diabetic *Stra6^-/-^* mice. (**H**,**I**) Fundus and fluorescence images for non-diabetic *Stra6^-/-^* mice. (**J**) Representative OCT mage for diabetic *Stra6^-/-^* mice. (**H**,**I**) Fundus and fluorescence images for diabetic *Stra6^-/-^* mice. (**M**) The outer nuclear layer thickness was measured from each genotype and condition (*n* = 4) and blotted in the graph. INL, inner nuclear layer; ONL, outer nuclear layer; and RPE, retina pigment epithelium.

## Data Availability

The authors confirm that the data supporting the findings of this study are contained within the article. The raw data are available upon request from the corresponding author.

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
