# Peer review of "Diabetes Aggravates Photoreceptor Pathologies in a Mouse Model for Ocular Vitamin A Deficiency"

_antioxidants, 2022, doi:10.3390/antiox11061142_

Round 1

Reviewer 1 Report

Comments to author

Ramkumar and coauthors, investigated the relationship between photoreceptors, Vitamin A and diabetes, using the STRA6-deficient animal model, divided in two main groups (WT and stra6-/-, and in to two subgroups: non-diabetic and diabetic). The Authors (AUs) observed that photoreceptors positively contribute to the Diabetic retinopathy progression. AUs concluded that, there is a significant correlation between Vitamin A metabolism and Diabetes, indicating an altered retinoid homeostasis in the eye. The severity of diabetes correlates positively to the decrease of circulating Vitamin A.  Their manuscript is an original and well written report. However, these concepts are not new, the Authors provide some new additional information to the scientific literature, already published in this medical field.

1. the data support the conclusions.

2. However, it contains misunderstandings and little typing errors, the article is well written and I recommend the authors to revise the article before resubmission.

minor concerns

page 4 line 147 change ANNOVA with ANOVA and add specifications of the GraphPad software.

page 4 line 157 remove the repeated word “membrane”.

page 4 line 163 specification of RPE65 missing.

page 4 insert the Statistic paragraph.

page 5 ELISA method is missing in the Material and Method section

page 6 line 228 remove space between of and vitamin A

page 6 line 232 remove space between stra6-/- and mice

page 9 line 311 remove space between stra6-/- and mice

page 11 line 363 remove space between WT mice. And (D).

page 11 line 365 remove space between for and non-diabetic

M&M section: please specify the number of animals used per each experiment

Author Response

Dear Editor,

We thank you and the reviewers for the valuable comments and suggestions. Below we have addressed all critiques provided by the reviewers. In heeding the recommendations of the reviewers, we have made extensive revisions to the text and made additions to Figures, clarified Figure legends and changed the abstract. In some cases, text that lead to misunderstandings of the reviewers has been clarified.

Revisions are presented in blue text. We have uploaded both an unmarked copy of the revised manuscript and a copy of the manuscript in which we have indicated changes by blue text to facilitate review.

We wish to thank you and the reviewers again for their efforts that helped us to prepare an improved revision. We hope that the manuscript is now acceptable for publication in the Journal.

Sincerely,

Ramkumar Srinivasagan & Johannes von Lintig

Reviewer_1

Ramkumar and coauthors, investigated the relationship between photoreceptors, Vitamin A and diabetes, using the STRA6-deficient animal model, divided in two main groups (WT and stra6-/-, and in to two subgroups: non-diabetic and diabetic). The Authors (AUs) observed that photoreceptors positively contribute to the Diabetic retinopathy progression. AUs concluded that, there is a significant correlation between Vitamin A metabolism and Diabetes, indicating an altered retinoid homeostasis in the eye. The severity of diabetes correlates positively to the decrease of circulating Vitamin A. Their manuscript is an original and well written report. However, these concepts are not new, the Authors provide some new additional information to the scientific literature, already published in this medical field.

  1. the data support the conclusions.
  2. However, it contains misunderstandings and little typing errors, the article is well written and I recommend the authors to revise the article before resubmission.

Answer: We thank the reviewer #1 for the overall positive evaluation. Please find the changes we made listed below:

minor concerns

page 4 line 147 change ANNOVA with ANOVA and add specifications of the GraphPad software.

Answer: We made this change and also indicate the version of the software.

page 4 line 157 remove the repeated word “membrane”.

Answer: We removed the repeated word.

page 4 line 163 specification of RPE65 missing.

Answer: We indicate the source of the antibody.

page 4 insert the Statistic paragraph.

Answer: We inserted as advised. (Page 5; line 191-195)

page 5 ELISA method is missing in the Material and Method section

Answer:  The HbA1c measurement was carried out using the Crystal Chem (Elk Grove Village, IL, USA) Mouse A1c kit. We followed the manufacturer’s protocol to perform this assay. This information was included in the revised manuscript (page-3: line 101).

page 6 line 228 remove space between of and vitamin A

Answer: We inserted as advised.

page 6 line 232 remove space between stra6-/- and mice

Answer: We removed as advised.

page 9 line 311 remove space between stra6-/- and mice

Answer: We removed as advised.

page 11 line 363 remove space between WT mice. And (D).

Answer: We removed as advised.

page 11 line 365 remove space between for and non-diabetic

Answer: We removed as advised.

M&M section: please specify the number of animals used per each experiment

Answer: Thank you for making us aware of this. At the beginning of the study, we used 12 male mice for each genotype and condition (diabetic and non-diabetic). After 8 weeks of diabetes, analyzed all mice for serum HBA1c. We also sacrificed 6 mice per genotype and condition for further analysis. After 30 weeks of diabetes, we sacrificed the remaining mice and analyzed them for the described parameters. For details of the different analyses, the numbers are provided with the figure legends.

Reviewer 2 Report

The manuscript by Ramkumar and colleagues seeks to investigate the interplay between diabetes and Vitamin A homeostasis in mice. The authors induce diabetes in control WT and ocular Vitamin A-compromised Stra6 knockout mice by streptozotocin (STZ) injection and then compare the single or combined effects of diabetes and ocular Vitamin A deficiency on retinoids concentration in the liver and eye, oxidative stress, rod and cone gene expression, retinal morphology and the ocular vasculature. They find that early (8 weeks) after the induction of diabetes, serum retinyl esters and liver retinol levels are significantly elevated in both diabetic WT and Stra6-/- mice compared to their respective controls. However, ocular retinoid homeostasis is affected by diabetes only in Stra6-/- mice, paradoxically resulting in a slight elevation towards WT levels. Both diabetes and Vitamin A deficiency result in increased oxidative stress as measured by superoxide production in the retina. Evaluation of the long-term effects of diabetes (after 30 weeks), confirmed the liver retinol findings from the early diabetic point. While total ocular retinoid was still reduced in Stra6-/- eyes, diabetic animals no longer showed improvement. Vitamin A deficiency was also associated with reduction in the expression of rod and cone opsin transducin genes, which was further exacerbated by diabetes in the case of rhodopsin, M-cone opsin, and rod and cone transducin. This protein loss was associated with mild degeneration observed in the Stra6-/- retinas. Overall, the authors show convincingly that diabetes affects retinoid homeostasis and aggravates the long-term consequences of Vitamin A deficiency. Notably, however, the bigger effect appears to be on the expression of photoreceptor proteins, likely caused by a combination of chromophore deficiency and retinal degeneration.

Overall, this is a rigorous and timely study that presents novel information that will be of interest to researchers in the fields of diabetes, Vitamin A homeostasis, oxidative stress and retinoid metabolism, and retinal biology. The Introduction is both scholarly and informative and does an excellent job introducing the relevant background information and the relevant outstanding questions in the field that this study is trying to address. The experiments are well constructed and their results are described clearly. The figures are generally easy to follow and inspire confidence in the rigor of the data. The Discussion is to the point and brings up several interesting points. In particular, this reviewer is excited about the future experiments with Vitamin A supplementation in diabetic subjects. Several concerns reduce enthusiasm for this otherwise excellent manuscript.

Major:

-          Figure 7 shows analysis of the expression of several visual cycle genes in control and diabetic WT mice but surprisingly/disappointingly does not present similar data for the more interesting case of Stra6-/- mice.

-          Oxidative stress (superoxide production) does not seem to have been determined in 30-week diabetic mice, which reduces the impact of the related discussion of the role of oxidative stress in diabetes.

-          Photoreceptor physiology is discussed in the context of thinning retina and reduced photoreceptor opsin and transducin expression levels but no direct measurements of retinal physiology (ERGs) are presented.

Minor:

-          The abstract is difficult to follow and does not make clear the key findings from the study in its current form. Consider revising to simplify and clean up to help readers grasp the key findings from the study and their implications.

-          Line 70: “interactions of diabetes vitamin A deficiency” should likely be “interactions of diabetes and vitamin A deficiency”

-          Please define eWAT in the Methods (line 95) or in the legend of Figure 4 (line 254).

-          Line 353: “was also observable in the superior part of the retina” should probably read “was also observable in the inferior part of the retina” although this is difficult to determine (see next point).

-          Please change the symbols in Figure 8M to make it easier to discriminate individual data sets.

Author Response

Dear Editor,

We thank you and the reviewers for the valuable comments and suggestions. Below we have addressed all critiques provided by the reviewers. In heeding the recommendations of the reviewers, we have made extensive revisions to the text and made additions to Figures, clarified Figure legends and changed the abstract. In some cases, text that lead to misunderstandings of the reviewers has been clarified.

Revisions are presented in blue text. We have uploaded both an unmarked copy of the revised manuscript and a copy of the manuscript in which we have indicated changes by blue text to facilitate review.

We wish to thank you and the reviewers again for their efforts that helped us to prepare an improved revision. We hope that the manuscript is now acceptable for publication in the Journal.

Sincerely,

Ramkumar Srinivasagan & Johannes von Lintig

Reviewer_2

The manuscript by Ramkumar and colleagues seeks to investigate the interplay between diabetes and Vitamin A homeostasis in mice. The authors induce diabetes in control WT and ocular Vitamin A-compromised Stra6 knockout mice by streptozotocin (STZ) injection and then compare the single or combined effects of diabetes and ocular Vitamin A deficiency on retinoids concentration in the liver and eye, oxidative stress, rod and cone gene expression, retinal morphology and the ocular vasculature. They find that early (8 weeks) after the induction of diabetes, serum retinyl esters and liver retinol levels are significantly elevated in both diabetic WT and Stra6-/- mice compared to their respective controls. However, ocular retinoid homeostasis is affected by diabetes only in Stra6-/- mice, paradoxically resulting in a slight elevation towards WT levels. Both diabetes and Vitamin A deficiency result in increased oxidative stress as measured by superoxide production in the retina. Evaluation of the long-term effects of diabetes (after 30 weeks), confirmed the liver retinol findings from the early diabetic point. While total ocular retinoid was still reduced in Stra6-/- eyes, diabetic animals no longer showed improvement. Vitamin A deficiency was also associated with reduction in the expression of rod and cone opsin transducin genes, which was further exacerbated by diabetes in the case of rhodopsin, M-cone opsin, and rod and cone transducin. This protein loss was associated with mild degeneration observed in the Stra6-/- retinas. Overall, the authors show convincingly that diabetes affects retinoid homeostasis and aggravates the long-term consequences of Vitamin A deficiency. Notably, however, the bigger effect appears to be on the expression of photoreceptor proteins, likely caused by a combination of chromophore deficiency and retinal degeneration.

Overall, this is a rigorous and timely study that presents novel information that will be of interest to researchers in the fields of diabetes, Vitamin A homeostasis, oxidative stress and retinoid metabolism, and retinal biology. The Introduction is both scholarly and informative and does an excellent job introducing the relevant background information and the relevant outstanding questions in the field that this study is trying to address. The experiments are well constructed and their results are described clearly. The figures are generally easy to follow and inspire confidence in the rigor of the data. The Discussion is to the point and brings up several interesting points. In particular, this reviewer is excited about the future experiments with Vitamin A supplementation in diabetic subjects. Several concerns reduce enthusiasm for this otherwise excellent manuscript.

Answer: We thank the reviewer for the generous evaluation and the rigorous review of the manuscript.

Major:

Figure 7 shows analysis of the expression of several visual cycle genes in control and diabetic WT mice but surprisingly/disappointingly does not present similar data for the more interesting case of Stra6-/- mice.

Answer: We employed 12 male mice for each genotype and condition (diabetic and non-diabetic). After 8 weeks of diabetes, we analyzed all mice for serum H1bA1c. We also sacrificed 6 mice per genotype and condition for further analyzes. After 30 weeks of diabetes, we sacrificed the remaining mice and analyzed them for the described parameters. For details of the different analyses, the numbers are provided with the figure legends. We had an extra-cohort of WT mice that allowed us to measure visual cycle protein levels by WB. We thought to include the data in the study because it focused on the effect of diabetes on vitamin A metabolism in the WT rather than studying the effects of diabetes on the mutant mice. Additionally, we thought that this measurement is clinically less significant because of the severe phenotype of STRA6 mutation in human patients.

Oxidative stress (superoxide production) does not seem to have been determined in 30-week diabetic mice, which reduces the impact of the related discussion of the role of oxidative stress in diabetes.

Answer: Retinal oxidative stress is a significant indicator of early diabetes. Therefore, we determined it after 8 weeks of diabetes. We focused on other hallmarks of diabetes in older mice.

Photoreceptor physiology is discussed in the context of thinning retina and reduced photoreceptor opsin and transducin expression levels but no direct measurements of retinal physiology (ERGs) are presented.

Answer: Several studies showed that ERG responses are diminished in diabetic WT mice. We showed that scotopic and photopic ERG responses are largely diminished in Stra6-/- mice (see Ref. 12). We decided to measure other parameters relevant to diabetes because of the limited number of mice in this study.

Minor:

The abstract is difficult to follow and does not make clear the key findings from the study in its current form. Consider revising to simplify and clean up to help readers grasp the key findings from the study and their implications.

Answer: We revised the abstract to better express the study’s purpose and endpoints.

Line 70: “interactions of diabetes vitamin A deficiency” should likely be “interactions of diabetes and vitamin A deficiency”

Answer: Thank you for pointing it out. We corrected this mistake.

Please define eWAT in the Methods (line 95) or in the legend of Figure 4 (line 254).

Answer: We defined eWAT as epididymal white adipose tissue in the method section (page-3 line 111)

Line 353: “was also observable in the superior part of the retina” should probably read “was also observable in the inferior part of the retina” although this is difficult to determine (see next point).

Answer: Thank you for pointing it out. We replaced superior by inferior.

Please change the symbols in Figure 8M to make it easier to discriminate individual data sets.

Answer: We display the symbols in color for better optical distinction.

Round 2

Reviewer 2 Report

The authors have addresses almost all of my initial concerns. Although I still believe that including data on the expression of visual cycle genes in control and diabetic Stra6-/- mice would have been informative, I understand that performing that experiment is not feasible at this time.